



# A global dataset of soil organic carbon mineralization under various incubation conditions

Shuai Zhang[1,2,3], Mingming Wang[1,2,3], Jinyang Zheng[1,2,3], Zhongkui Luo[1,2,3]

[1]State Key Laboratory of Soil Pollution Control and Safety, Zhejiang University, Hangzhou 310058, China
[2]Key Laboratory of Environment Remediation and Ecological Health, Ministry of Education, Zhejiang University, Hangzhou 310058, China
[3]Zhejiang Key Laboratory of Agricultural Remote Sensing and Information Technology, College of Environmental and Resource Sciences, Zhejiang University, Hangzhou 310058, China

*Correspondence to*: Zhongkui luo (luozk@zju.edu.cn)

**Abstract.** Microbial decomposition of soil organic carbon (SOC) is a major source of atmospheric $CO_2$ and a key component of climate-carbon feedbacks. Understanding how SOC mineralization responds to temperature is essential for improving climate projections. Here, we compiled a global dataset of laboratory incubation experiments measuring SOC mineralization across diverse soils and temperature regimes. The dataset reveals that 84% of samples originated from surface soils (0–30 cm), and 50% of incubations lasted fewer than 50 days. Incubation temperatures ranged from –10 to 60 °C, with temperature
intervals used to estimate temperature sensitivity ($Q_{10}$) spanning 2–40 °C; notably, 81% of $Q_{10}$ estimates were based on intervals exceeding 5 °C. Moreover, in 61% of cases, the lower incubation temperature for $Q_{10}$ estimation differed from the mean annual temperature at the sampling site by more than 5 °C, indicating a mismatch with in situ conditions. Our analysis highlights critical gaps in current experimental designs, particularly the underrepresentation of subsoils (>30 cm) and the use of temperature ranges that deviate from field conditions. We further evaluated the ability of 16 temperature response functions
used in 69 Earth System Models to capture SOC mineralization patterns. Most models failed to reproduce empirical temperature response, especially at higher temperatures, albeit multi-term exponential functions showed relatively better performance. By coupling our dataset with a two-pool carbon model, we found that external environmental constraints and the intrinsic temperature response (including SOC decomposability and microbial processes) similarly influence the temperature sensitivity of SOC mineralization at the global scale, with their relative importance varying across ecosystem types. Our
findings underscore the need for incubation experiments that better represent field conditions—both in depth and temperature range—and call for improved model parameterizations to enhance SOC feedback projections under future climate scenarios. The dataset is archived and publicly available at https://doi.org/10.6084/m9.figshare.25808698 (Zhang et al., 2025)

## 1 INTRODUCTION

Soils annually release approximately five times more $CO_2$-C to the atmosphere via microbial mineralization of soil organic
carbon (SOC) than all anthropogenic fossil fuel emissions combined (Tang et al., 2020). As a key flux in the global carbon



cycle, this soil-derived $CO_2$ efflux is projected to intensify under global warming (Lei et al., 2021; Wang et al., 2022) due to the inherent temperature sensitivity of microbial decomposition (Davidson & Janssens, 2006). Yet, the magnitude and mechanisms of this feedback remain contentious (Crowther et al., 2016; Soong et al., 2021), posing a critical uncertainty in Earth System Models (ESMs) projections of future climate-carbon dynamics.

The temperature sensitivity of SOC mineralization is commonly expressed $Q_{10}$—the factor by which the mineralization rate increases for every 10 °C rise in temperature. $Q_{10}$ is typically calculated following Eq.(1):

$$Q_{10} = \left(\frac{R_{T_2}}{R_{T_1}}\right)^{\frac{10}{T_2-T_1}}, \tag{1}$$

where $R_{T_1}$ and $R_{T_2}$ are the SOC mineralization (often microbial respiration) rates at low temperature ($T_1$) and high ($T_2$) temperatures, respectively. Most ESMs adopt a constant or temperature-dependent $Q_{10}$ value (Luo et al., 2016; Luo, Luo,

Wang, Xia, & Peng, 2020), but empirical $Q_{10}$ estimates vary widely due to numerous influencing factors (Haaf, Six, & Doetterl, 2021; Patel et al., 2022), including calculation approaches (Hamdi, Moyano, Sall, Bernoux, & Chevallier, 2013), environmental constraints such as soil pH (Craine, Fierer, & McLauchlan, 2010) and clay content (Hartley, Hill, Chadburn, & Hugelius, 2021), climatic conditions like precipitation (Li, Pei, Pendall, Fang, & Nie, 2020), and microbial community traits (Wang et al., 2021). These controls can be grouped into three primary mechanisms: (1) Carbon pool quality: the chemical

composition of SOC influences its thermodynamic properties and decomposability (Haddix et al., 2011); (2) Microbial community structure and function: Variations in microbial traits affect SOC decomposition efficiency and enzyme production (Karhu et al., 2014; Xiao et al., 2023); and (3) Physicochemical protection and accessibility: Soil texture, aggregation, and mineral interactions modulate the accessibility of SOC to microbial enzymes (Gershenson, Bader, & Cheng, 2009). While these mechanisms are often discussed independently, their relative contributions and interactions remain poorly understood at

the global scale (Jones, Cox, & Huntingford, 2003).

Temperature sensitivity is typically assessed via either field or laboratory incubation experiments. Field studies reflect in situ conditions but are confounded by numerous environmental variables (e.g., plant inputs, soil moisture variability), and it is difficult to separate root and microbial respiration. Moreover, field measurements are challenging to conduct continuously, especially in remote ecosystems. Laboratory incubations, while simplified and often subject to preparation artifacts (e.g.,

sieving, drying, rewetting), offer controlled conditions that isolate specific mechanisms and allow for systematic comparisons across soils and temperatures (Zhang, Yu, Lin, & Zhu, 2020). Importantly, although many laboratory studies have yielded mechanistic insights, they are often limited in spatial scope or designed to test specific hypotheses. Yet, taken together, the body of global incubation data represents an underutilized resource for addressing broad-scale questions about SOC temperature sensitivity.

Here, we compile and synthesize a global dataset of time-series measurements of SOC mineralization under controlled laboratory incubation conditions, encompassing diverse soil types, climatic zones, and incubation protocols. The dataset is valuable for characterizing SOC mineralization processes and their response to temperature in relation to various soil properties and incubation conditions. Using this dataset, we evaluate the performance of temperature response functions currently used





in ESMs against observed $Q_{10}$ data, and use a two-pool carbon model to simulate SOC mineralization and assess the relative

influence of different regulatory mechanisms on temperature sensitivity. By integrating empirical observations with process-based modeling, our study provides mechanistic insights into the drivers of SOC temperature sensitivity and informs efforts to improve Earth system model projections under climate change.

## 2 THE DATA

We compiled a global dataset of laboratory incubation experiments to investigate the temperature sensitivity of SOC

mineralization. Literature searches were conducted using the Web of Science and the Chinese National Knowledge Infrastructure (CNKI). The search terms included:

*"soil AND (respir\* OR ((carbon OR $CO_2$ OR carbon dioxide OR organic matter) AND (flux OR efflux OR emission OR release OR loss OR mineraliz\* OR decompos\*))) AND (temperature OR warm\* OR cool\*) AND incubat\*"*

In addition to dataset queries, we screened all studies cited in five previous synthesis papers on temperature sensitivity of SOC

mineralization (Fierer, Colman, Schimel, & Jackson, 2006; Hamdi et al., 2013; Ren et al., 2020; Christina Schädel et al., 2020; Wang et al., 2019). To be included in our dataset, studies had to meet the following criteria:

  1) The incubated soil must be sampled from the mineral layer;

  2) Each experiment must incubate the same soil at two or more temperatures;

  3) All other incubation conditions (e.g., moisture) must be identical across temperature treatments and maintained

throughout the incubation; and

  4) Time-series data of carbon mineralization rates or cumulative carbon mineralization must be reported.

Using these criteria, we identified 191 publications, encompassing 721 distinct soils and totaling 21,979 data points on SOC mineralization (Fig. 1).

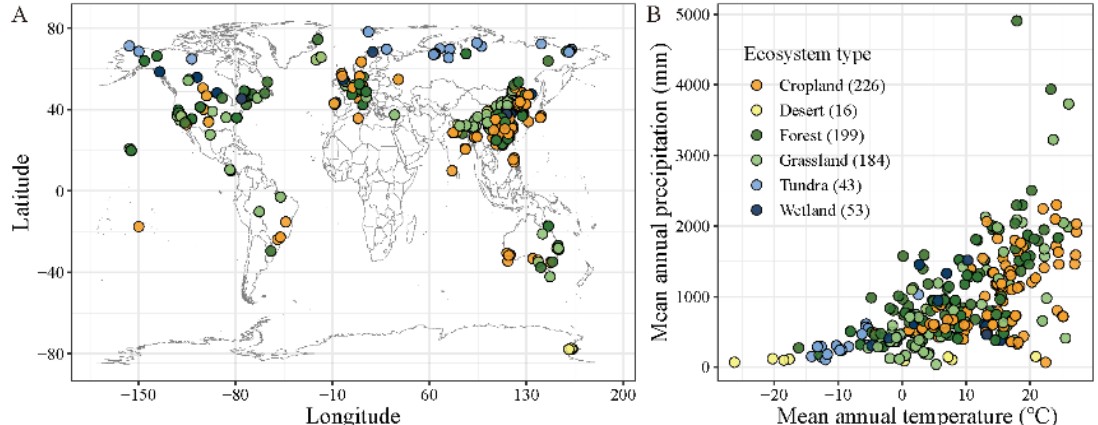

**Figure 1: Geographic distribution of soil samples.** A, soil sample locations; B, distribution across climate conditions and ecosystems. Numbers in parentheses show the sample size in the specified ecosystem.



When available, numerical data were directly extracted from the publications, and graphical data were digitized using the WebPlotDigitizer (Burda, O'Connor, Webber, Redmond, & Perdue, 2017). SOC mineralization rates were standardized to g $CO_2$-C kg$^{-1}$ SOC d$^{-1}$ and g $CO_2$-C kg$^{-1}$ soil d$^{-1}$. Cumulative mineralization was also recorded as g $CO_2$-C kg$^{-1}$ SOC and g $CO_2$-C kg$^{-1}$ soil, corresponding to the total mineralized carbon over the duration of the incubation. We also compiled ancillary information when available, including soil properties [e.g., pH, total nitrogen (TN), carbon-to-nitrogen ratio (C:N), soil bulk density (BD), and texture)], site characteristics (geographic coordinates and ecosystem type), and experimental design (incubation temperature, duration, moisture condition, and pretreatment) (Table 1). Based on the recorded geographic coordinates of sampling locations, we extracted 19 climate variables from WorldClim V2.0 at a spatial resolution of 1 km$^2$ (Fick & Hijmans, 2017). All complied data are deposited to https://doi.org/10.6084/m9.figshare.25808698 (Zhang et al., 2025) and are publicly accessible.

**Table 1: Variables included in the dataset.**

| Variable | Description | Units |
| --- | --- | --- |
| Publication information | | |
| Reference_ID | Reference ID | - |
| First_author | First author of the publication | - |
| Publication_year | Publication year | Year |
| Sampling_year | Sampling year | Year |
| Journal | Journal name of the publication | - |
| Title | Title of the article | - |
| Site information | | |
| Latitude | Latitude, positive = North, negative = South | Decimal |
| Longitude | Longitude, positive = East, negative = West | Decimal |
| MAT | Mean annual temperature, extracted from WorldClim 2.1 based on the latitude and longitude of soil sampling sites, the data is the 30-year mean value from 1970 to 2000 | °C |
| MAP | Mean annual precipitation, extracted from WorldClim 2.1 based on the latitude and longitude of soil sampling sites, the data is the 30-year mean value from 1970 to 2000 | mm |
| Elevation | Elevation, extracted from WorldClim 2.1 based on the latitude and longitude of soil sampling sites | m |
| Eco_type | Ecosystem type (grassland, forest, etc.) | - |
| Species | The aboveground plant species at the sampling site | - |
| Soil_ID | Soil ID | - |
| Profile_ID | Profile ID | - |
| Soil_depth | The top and bottom depths of the sampled soil (0_10, 0_20, etc.). Some studies only provide the horizon of the soil profile, such as A | cm |



| | horizon, B horizon | |
|---|---|---|
| **Soil characteristics** | | |
| SOC | Initial soil organic carbon content | % |
| TN | Initial soil total nitrogen content | % |
| C:N | Soil carbon:nitrogen ratio | - |
| pH | Initial soil pH | - |
| BD | Soil bulk density | $g \cdot cm^{-3}$ |
| Soil texture | Clay, silt, and sand | % |
| **Incubation information** | | |
| Incu_duration | Incubation duration | Day |
| Incu_temp | Incubation temperature | °C |
| Soil_mass | The dry weight of the incubated soil | g |
| C_input | Carbon input at the beginning of incubation (biochar, glucose, etc.) | - |
| Input_amount | The amount of carbon input at the beginning of the incubation expressed as a percentage of the initial soil organic carbon content | % |
| Measure_day | Measurement day for carbon mineralization | Day |
| FC | Soil moisture content is expressed as a percentage of field capacity (e.g., 60% FC indicates 60% of the maximum field capacity). | % |
| Gravity | Soil gravity water content | % |
| Pre_incubation | Pre-incubation duration | Day |
| Pre_treatment | Pre-treatment before the beginning of the incubation (e.g., fresh homogenized, air-dried, etc.) | - |
| Sieve | The sieving size prior to the beginning of the incubation | mm |
| CO$_2$_method | Determination method of mineralized $CO_2$, including gas chromatograph, alkali absorption, and infrared gas analysis | - |
| Exp_ID | Experiment ID. The same ID includes mineralization data of the same soil at different incubation temperatures, and other incubation conditions were identical at different temperatures | - |
| n | Number of replicates of a incubation | - |
| **Mineralization information** | | |
| Rate_soil | Time-course carbon mineralization rate, which was normalized to per kilogram of soil | $mg\ CO_2\text{-}C\ kg^{-1}\ soil\ d^{-1}$ |
| SD_rate_soil | The standard deviation of Rate_soil | $mg\ CO_2\text{-}C\ kg^{-1}\ soil\ d^{-1}$ |
| Rate_SOC | Time-course carbon mineralization rate, which was normalized to per kilogram of SOC | $g\ CO_2\text{-}C\ kg^{-1}\ SOC\ d^{-1}$ |
| SD_rate_SOC | The standard deviation of Rate_SOC | $g\ CO_2\text{-}C\ kg^{-1}\ SOC\ d^{-1}$ |
| Cumu_soil | Time-course cumulative carbon mineralization, which was | $mg\ CO_2\text{-}C\ kg^{-1}\ soil$ |



| | normalized to per kilogram of soil | |
|---|---|---|
| SD_cumu_soil | The standard deviation of Cumu_soil | mg $CO_2$-C $kg^{-1}$ soil |
| Cumu_SOC | Time-course cumulative carbon mineralization, which was normalized to per kilogram of SOC | g $CO_2$-C $kg^{-1}$ SOC |
| SD_cumu_SOC | The standard deviation of Cumu_SOC | g $CO_2$-C $kg^{-1}$ SOC |

## 3 INSIGHTS FROM THE DATASET

### 3.1 Spatial coverage

Our dataset captures a broad global distribution of soil incubation experiments, with sampling sites concentrated in China, Europe, and the United States (Fig. 1A). However, samples are relatively sparse in Australia, Canada, and Russia, with almost absent in Africa. This geographic imbalance is particularly concerning given the importance of tropical and high-latitude cold regions for global carbon storage and their heightened vulnerability to climate change. Addressing these data gaps is critical for improving the accuracy of global SOC-climate feedback projections.

The dataset covers major terrestrial ecosystems (Fig. 1B), including croplands (226 sites), forests (199), and grasslands (184), but includes relatively few samples from tundra (43), wetlands (53), and deserts (16). Yet, tundra and wetland soils are known for their high SOC content and may exhibit distinct temperature responses due to unique environmental conditions (Wang et al., 2022). In tundra ecosystems, SOC is dominated by particulate organic carbon, which is more sensitive to warming than mineral-associated organic carbon (Georgiou et al., 2024). Moreover, freeze-thaw cycles can disrupt microbial and

physical protection mechanisms, altering SOC turnover (Schuur et al., 2009). Similarly, wetland soils experience fluctuating redox conditions driven by water table changes, potentially leading to nonlinear SOC responses to warming (Wang, Wang, He, & Feng, 2017). These complexities reinforce the need for targeted studies in underrepresented ecosystems.

SOC content in the dataset ranges from 0.04% to 58.85%, with a median of 2.48% (Fig. 2A). Notably, 73% of samples contain less than 5% SOC, with higher values mostly occurring in wetland soils. Incubation temperatures range from –10 to

60 °C, with a median of 17 °C and frequent use of standard temperatures such as 5 °C, 15 °C, and 25 °C (Fig. 2B). $Q_{10}$ values, calculated from paired temperature treatments, are most derived from 15–25 °C (Fig. 2E), with 10 °C temperature difference (i.e., $\Delta T$, the difference between $T_2$ and $T_1$ in equation 1) accounting for 34 % of cases (Fig. 2F). However, only 19% of experiments used $\Delta T \leq 5$ °C, a range more reflective of projected climate warming (IPCC, 2023).

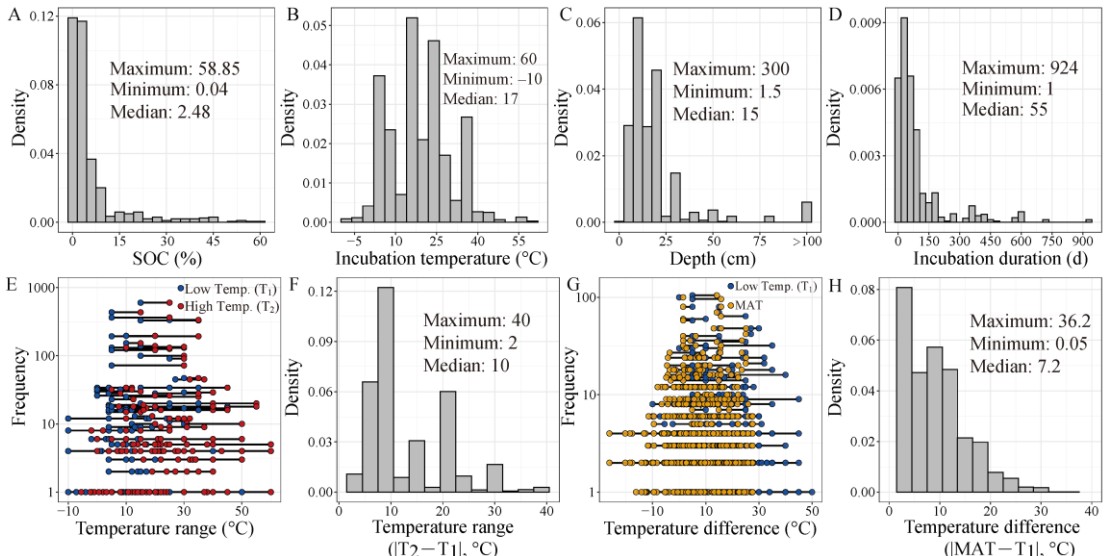

**Figure 2: Basic characteristics of the incubation dataset.** The distribution (frequency or density) of soil organic carbon content (A), incubation temperature (B), soil sampling depth (C), incubation duration (D), temperature range ($T_1$ and $T_2$ in equation 1) used for $Q_{10}$ estimation (E), absolute temperature range ($|T_2 - T_1|$) (F), temperature difference between the low incubation temperature ($T_1$ in equation 1) and the local mean annual temperature (MAT) at the sampling site (G), absolute temperature difference ($|MAT - T_1|$) (H). In panels E and G, the blue circles represent the low incubation temperature ($T_1$ in equation 1), the red circles indicate the high incubation temperature ($T_2$ in equation 1), and the yellow circles correspond to the mean annual temperature at the sampling site. Note the $\log_{10}$ scale of the $y$-axis in panels E and G. Most of the data points in panel E fall within the temperature ranges of 15-25 °C, 5-15 °C, 5-25 °C, 15-35 °C, and 25-35 °C.

## 3.2 Incubation temperature

While laboratory incubations allow precise control of environmental variables, their ecological relevance depends critically on the selection of incubation temperatures. SOC mineralization often responds nonlinearly to warming (Melillo et al., 2017), especially in cold ecosystems where small temperature increases can trigger large $CO_2$ emissions (Turetsky et al., 2020). However, many studies apply large $\Delta T$ values (>10 °C), which may obscure subtle thresholds, suppress key microbial feedbacks, and limit the transferability of findings to field conditions.

This limitation is compounded by the mismatch between incubation temperature and field conditions. We compared the low incubation temperature (i.e., $T_1$ in equation 1) used for estimation to the local mean annual temperature (MAT) at each sampling sites (Fig. 2G). In 61% of the cases, the absolute difference between $T_1$ and MAT exceeded 5 °C (Fig. 2H), potentially biasing $Q_{10}$ estimates, as temperature sensitivity is itself temperature-dependent (Alster et al., 2023; Hamdi et al., 2013; Patel et al., 2022). To enhance ecological validity, we recommend future studies align incubation temperatures more closely with local MATs, particularly when estimating $Q_{10}$.

Most soil samples were collected from surface layers: 84% originate from the 0–30 cm depth (Fig. 2C). However, subsoils (>30 cm) store more than twice the SOC of topsoil globally (Jobbágy & Jackson, 2000), and emerging evidence suggests they are not inert, but can respond sensitively to warming (Hicks Pries, Castanha, Porras, & Torn, 2017; Hicks Pries et al., 2023). SOC dynamics in deeper layers are governed by different stabilization processes and environmental controls, including lower





oxygen availability, reduced root inputs, and greater mineral association (Jia et al., 2019; Xu et al., 2021). These vertical gradients shape SOC quality, microbial access, and thus, temperature sensitivity. Current underrepresentation of deep soils in
incubation experiments limits our ability to predict long-term carbon–climate feedbacks and highlights the need for deeper sampling in future work.

**3.3 Incubation duration**

Incubation durations vary widely across studies. While some experiments extend for several years, 80% of the incubations lasted <113 days, and half were <54 days (Fig. 2D). Short-term incubations are efficient and cost-effective, and are well suited
for capturing the dynamics of labile carbon pools that dominate initial $CO_2$ release (Schädel et al., 2020). They also minimize microbial adaptation and maintain more natural soil structure. However, they may overlook the slower dynamics of recalcitrant carbon pools, which contribute substantially to long-term SOC persistence and climate feedbacks (Schmidt et al., 2011).

In contrast, long-term incubations are essential for capturing the decomposition of slow-cycling SOC fractions, especially in the absence of new carbon inputs. As labile carbon is depleted, persistent carbon pools increasingly dominate respiration,
providing insights into intrinsic SOC stability (Schädel et al., 2020). Long-term studies also enable assessment of microbial community shifts and potential feedbacks under sustained warming (Guan et al., 2022; Jerry M Melillo et al., 2017). Yet, they also introduce new complexities, including potential changes in soil structure, microbial acclimation, and moisture loss, which may confound temperature effects (Kirschbaum, 2006). We advocate for a combined approach that integrates both short- and long-term incubations. This dual strategy can capture early-stage microbial dynamics, as well as long-term decomposition
pathways of stable carbon pools. By leveraging both timescales, researchers can better disentangle microbial versus physiochemical controls and derive more robust parameter estimates for Earth system models.

**4 COMPARISON WITH TEMPERATURE RESPONSE FUNCTIONS**

Earth System Models (ESMs) are key tools for projecting SOC dynamics under climate change, yet their predictive accuracy hinges on the reliability of temperature response functions for SOC mineralization. We examined 69 ESMs used in the Coupled
Model Inter-comparison Project Phase 6 (CMIP6) and identified 16 distinct temperature response functions (Fig. 3; Table 2 and Table S1). These functions differ markedly in structure, particularly at temperatures above 20 °C, where predicted mineralization rates diverge substantially (Fig. 3A). Most functions are empirical in nature and fall into four broad categories (Table 2):

1) Simple exponential models – assume fixed temperature sensitivity (e.g., constant $Q_{10}$ or classical Arrhenius);
2) Flexible $Q_{10}$ models – allow $Q_{10}$ to vary with temperature, typically through parameterized functions;

3) Non-linear empirical models – capture physiological thresholds, saturation effects, or inhibition at high temperatures; and



4) Hybrid/adjusted exponential models – incorporate additional terms to improve empirical fits (e.g., multi-term exponential or polynomial-exponential hybrids).

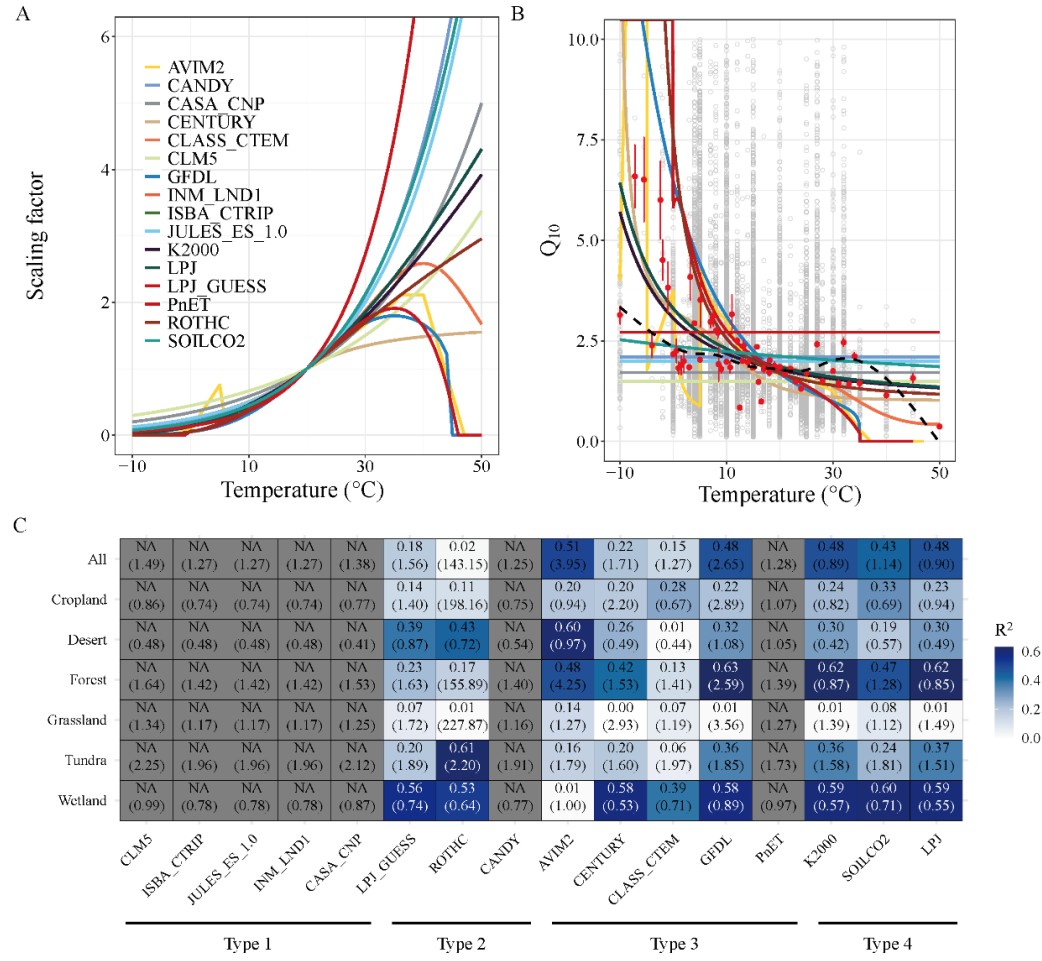


**Figure 3: Performance of soil carbon temperature response functions.** Temperature response functions of soil organic carbon mineralization (A), temperature sensitivity ($Q_{10}$) predicted by the temperature response functions (B), and function performance by comparing function predicted and observed $Q_{10}$ (C). Gray open points in panel B represent observed $Q_{10}$ values at different temperatures, while the black dashed curve shows the best-fit relationship between $Q_{10}$ and temperature based on locally weighted polynomial regression.
Red points indicate the mean values under the corresponding temperatures, and error bars represent one standard error of the observations. Panel C compares the observed mean $Q_{10}$ values at different temperatures with the $Q_{10}$ values predicted by the temperature response functions, presented for both global average and subsets grouped by ecosystem type. In panel C, numbers outside the parentheses represent the coefficient of determination ($R^2$), while numbers inside the parentheses indicate the root mean square error. Grey grids represent cases where $R^2$ could not be calculated due to constant $Q_{10}$ values defined by the respective temperature response functions.

**Table 2: Temperature response functions used in 69 earth system models. $Q_{10\_mod}$ was estimated based on its definition, using the following equation: $Q_{10\_mod} = f(T + 10)/f(T)$.**

| Types | Temperature response functions | Land carbon | Land surface models | Modelling | Models | References |
|---|---|---|---|---|---|---|





|  |  |  |  |  | centre |  |
|---|---|---|---|---|---|---|
| 1 | $f(T) = 1.5^{\frac{T-25}{10}}$ | CLM5 | CLM5 | CESM | CESM2 | Emmons et al. (2020) |
|  | $f(T) = 2^{\frac{T-30}{10}}$ | ISBA-CTRIP | ISBA-CTRIP | CNRM | CNRM-ESM2-1 | Séférian et al. (2019) |
|  | $f(T) = 2^{\frac{T-25}{10}}$ | JULES-ES-1.0 | JULES-ES-1.0 | UK | UKESM1-0-LL | Good et al. (2019) |
|  | $f(T) = 2^{\frac{T-10}{10}}$ | INM-LND1 | INM-LND1 | INM | INM-CM5-0 | Volodin et al. (2017) |
|  | $f(T) = 1.71^{\frac{T-35}{10}}$ | CASA-CNP | CABLE2.4+CASA-CNP | CSIRO | ACCESS-ESM1.5 | Ziehn et al. (2019) |
| 2 | $f(T) = 0.0326 + 0.00351 \cdot T^{1.652} - \left(\frac{T}{41.748}\right)^{7.19}$ | LPJ-GUESS | LPJ-GUESS | EC-Earth | EC-Earth3-CC | Smith et al. (2014) |
|  | $f(T) = \dfrac{47.9}{1 + exp\left(\frac{106}{T + 18.3}\right)}$ | RothC | ROTHC | Rothamsted | RothC | Coleman and Jenkinson (1996) |
|  | $f(T) = \begin{cases} 2.1^{\frac{T-35}{10}}, & T \le 35 \\ 1.0, & T > 35 \end{cases}$ | CANDY | CANDY | ULHG | CANDY | Franko, Oelschlägel, and Schenk (1995) |
| 3 | $f(T) = \begin{cases} 0.01, & -5 \ge T \\ 0.04, & -5 < T \le 0 \\ 0.04 + 0.06 \cdot T, & 0 < T \le 5 \\ 0.07 + 0.016 \cdot (T-5), & 5 < T \le 10 \\ 0.15 + 0.03 \cdot (T-10), & 10 < T \le 35 \\ 0.95, & 35 < T \le 40 \\ 0.95 - 0.135 \cdot (T-40), & 40 < T \le 47 \\ 0, & 47 < T \end{cases}$ | AVIM2 | BCC-AVIM2 | BCC | BCC-CSM2-MR | Ji and Yu (1999) |
|  | $f(T) = 0.56 + 0.465$ $\cdot arctan\big(0.097 \cdot (T - 15.7)\big)$ | CENRUTY | CENRUTY | CSU | CENRUTY | Parton, Schimel, Cole, and Ojima (1987) |
|  | $f(T) = Q_{10}^{\frac{T-15}{10}}$ $Q_{10} = 1.44 + 0.56 \cdot tanh\big(0.075 \cdot (46 - T)\big)$ | CLASS-CTEM | CLASS-CTEM | CCCma | CanESM5 | Swart et al. (2019) |
|  | $f(T) = T_1^{0.2} \cdot T_2$ | GFDL | GFDL-ESMM2M | GFDL | GFDL-ESMM2M | Shevliakova et al. (2009) |
|  | $f(T) = 0.68 \cdot exp\big(0.1 \cdot (T - 7.1)\big)$ | PnET | PnET-CN | UNH | PnET-CN | Aber, Ollinger, and Driscoll (1997) |
| 4 | $f(T) = exp\left(3.36 \cdot \left(\frac{T - 40}{T + 46.05}\right)\right)$ | K2000 | K2000 | CSIRO | K2000 | Kirschbaum (2000) |
|  | $f(T) = exp\left(\frac{E \cdot (T - T_{20})}{R \cdot (273.15 + T) \cdot (273.15 + T_{20})}\right)$ $R = 8.314\, J\, K^{-1}\, mol^{-1},$ $E = 55.5\, kJ\, mol^{-1},$ $T_{20} = 20\,℃$ | SOILCO2 | SOILCO2 | USDA | SOILCO2 | Šimůnek and Suarez (1993) |
|  | $f(T) = exp\left(308.56 \cdot \left(\frac{1}{56.02} - \frac{1}{T + 46.02}\right)\right)$ | LPJ | MRI-LCCM2 | MRI | MRI-ESM-2.0 | Yukimoto et al. (2019) |



This diversity reflects both the absence of a mechanistic consensus on temperature sensitivity and the trade-offs between functional realism, parameter interpretability, and computational efficiency.

Across all model types, a consistent feature is the prediction of lower SOC mineralization rates at lower temperatures (Fig. 3A). This conforms with known biological constraints – low temperatures suppress microbial activity and freeze liquid water, thereby restricting substrate diffusion and microbial access. However, substantial uncertainty persists regarding mineralization responses at elevated temperatures. Specifically, the temperature response functions yield three distinct response patterns:

1) Monotonic increase – mineralization rates rise continuously with temperature (e.g., classic Arrhenius behaviour; (Fang,
Singh, Matta, Cowie, & Van Zwieten, 2017));

2) Plateau – mineralization rates increase to an asymptote beyond which additional warming has little effect; and

3) Peak followed by decline – mineralization rates increase to an optimum and then drop due to thermal inhibition of enzymes, microbial stress, or substrate exhaustion.

Empirical studies support all three behaviours under different contexts, highlighting the need for flexible models that can
accommodate nonlinearities and thresholds in warming responses (Alster et al., 2023).

To evaluate model performance, we calculated observed $Q_{10}$ ($Q_{10\_obs}$) from our global incubation dataset using Equation (1), and compared them to modelled $Q_{10}$ values ($Q_{10\_mod}$) derived from each function. $Q_{10\_obs}$ was computed for each experiment, then aggregated by incubation temperature ($T_1$) to derive the global mean $Q_{10\_obs}$ at each temperature. There were then compared to $Q_{10\_mod}$ at corresponding temperatures for each model function (Fig. 3B-C). The results indicate that $Q_{10\_obs}$ varies
widely but exhibits a nonlinear decline with increasing temperature (Fig. 3B), consistent with metabolic theory and enzyme kinetics (Gillooly, Brown, West, Savage, & Charnov, 2001). Among the 16 tested functions, Type 4 (hybrid/adjusted exponential) functions performed best, with $R^2$ values of > 0.4 and rooted mean square errors (RMSE) < 1.2 (Fig. 3C). Type 1 (simple exponential) functions ranked second in performance, while Type 2 (flexible $Q_{10}$) functions consistently underperformed ($R^2 < 0.2$). Notably, all functions performed adequately within the 10–30 °C range – where most $Q_{10\_obs}$ values
clustered around 2 – but were less reliable below 10 °C or above 30 °C (Fig. 3B), where sample sizes were limited and biological responses are less predictable.

Ecosystem-specific performance varied substantially (Fig. 3C and S1). In forest ecosystems, Type 4 and Type 3 functions performed best, capturing both the magnitude and temperature dependency of $Q_{10\_obs}$, while models such as CLASS-CTEM consistently underpredicted sensitivity. Wetland soils also showed good agreement with most functions (except AVIM2),
although the small number of observations and narrow temperature range warrant caution. In croplands and grasslands, $Q_{10\_obs}$ values remained relatively stable (~2) across all temperatures (Fig. S1), likely due to uniform substrate quality, frequent anthropogenic disturbances (e.g., tillage, fertilization), and homogenized microbial communities, which dampen temperature responsiveness. Accordingly, Type 1 models—emphasizing constant $Q_{10}$—performed best in these systems. However, data scarcity remains a limiting factor for evaluating model performance in tundra, desert, and high-latitude cold systems. These
ecosystems, while storing vast amounts of SOC and being highly sensitive to warming, remain underrepresented in both





incubation data and model calibration. Their unique dynamics—driven by freeze–thaw cycles, moisture constraints, and slow microbial turnover—may necessitate tailored temperature response formulations not currently embedded in most ESMs.

Taken together, our results highlight that: (1) no single temperature response function captures $Q_{10\_obs}$ variability across all ecosystems and temperature ranges; (2) simple and hybrid exponential functions show relatively robust performance, particularly in cropland, grassland, and forest soils; (3) high-latitude, subsoil, and high-temperature responses remain poorly constrained due to data limitations; (4) expanding observational datasets across diverse ecosystems—especially in extreme environments—is essential for improving the realism and generalizability of temperature response functions in ESMs. Ultimately, our comparison provides a benchmark for refining temperature sensitivity formulations in soil carbon models, emphasizing the need for ecosystem-specific calibration and incorporation of nonlinear temperature effects to reduce uncertainty in future carbon-climate feedback projections.

## 5 COMBINING THE DATA WITH CARBON MODELS

The extensive spatial and environmental coverage of our global SOC mineralization dataset offer a unique opportunity to explore the mechanisms regulating the temperature sensitivity of microbial decomposition. To fully harness this potential, mechanistically-informed modelling approaches are essential. Here, we integrate the dataset with commonly used pool-based carbon models to test the relative contributions of different regulatory mechanisms. SOC mineralization is controlled by both intrinsic and extrinsic factors. Intrinsic factors include the chemical decomposability of SOC pools and the thermal traits of microbial communities – collectively referred to as the inherent temperature response (Davidson & Janssens, 2006). These control the baseline temperature dependence of microbial activity and carbon use efficiency. In contrast, extrinsic factors – such as mineral associations, aggregate occlusion, moisture limitation, and oxygen availability – operate as external environmental constraints, restricting microbial access to otherwise decomposable organic matter (Dungait, Hopkins, Gregory, & Whitmore, 2012). Disentangling these two mechanisms is critical, as they operate on different scales and are likely to respond differently to climate change. We propose a modelling framework in which SOC temperature sensitivity is partitioned into these two components, allowing us to quantify their relative contributions (Fig. 4).





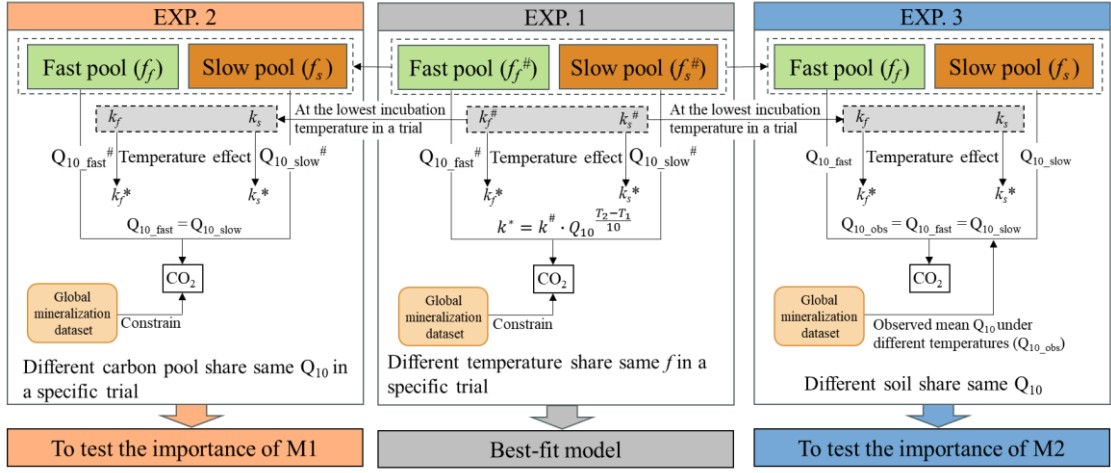

**Figure 4: The data-model integration framework to distinguish the importance of intrinsic temperature responses and external environmental constraints.** The framework illustrates the data-model fusion process employed to verify the regulatory mechanisms underlying the temperature sensitivity of soil organic carbon mineralization. $k_f$ and $k_s$ represent the decomposition rates (d$^{-1}$) of the fast ($f_f$) and slow ($f_s$) carbon pools, respectively. $Q_{10\_fast}$ and $Q_{10\_slow}$ are the temperature sensitivity of fast and slow carbon pools, respectively. In EXP. 1, $k_f$ and $k_s$ at the lowest temperature of each trial were optimized and then were scaled to other temperatures using $Q_{10\_fast}$ and $Q_{10\_slow}$, which were also optimized. The pool size ($f$) and decomposition rates ($k_f$ and $k_s$) at the lowest temperature from EXP. 1 were applied in EXP. 2 and EXP. 3.

## 5.1 A two-pool model of SOC mineralization

To represent SOC heterogeneity and its decomposition dynamics, we adopt a two-pool first-order model, distinguishing between fast- and slow-cycling carbon pools. The mineralization rate $R_t$ (g $CO_2$-C kg$^{-1}$ SOC d$^{-1}$) at time $t$ is expressed following Eq.(2):

$$R_t = k_f \cdot f_f \cdot C_0 \cdot e^{-k_f \cdot t} + k_s \cdot (1 - f_f) \cdot C_0 \cdot e^{-k_s \cdot t} , \qquad (2)$$

where $k_f$ and $k_s$ are the decomposition rate constants (d$^{-1}$) for the fast and slow pools, respectively; $f_f$ is the initial fraction of the fast pool in initial total SOC ($C_0$); $t$ is time in days. Temperature sensitivity is introduced via a $Q_{10}$ formulation following Eq.(3):

$$k_T = k_{ref} \cdot Q_{10}^{\frac{T - T_{ref}}{10}} , \qquad (3)$$

where $k_T$ is the decomposition rate of a carbon pool at incubation temperature $T$ (°C); $k_{ref}$ is the decomposition rate of the pool at a defined reference temperature ($T_{ref}$); $Q_{10}$ is the temperature sensitivity factor.

## 5.2 Simulation experiments

We conducted three simulation experiments to assess how well each regulatory mechanism explains the observed temperature sensitivity. In all experiments, model parameters were optimized by minimizing RMSE between observed and modeled SOC





mineralization rates using the DEoptim package in R4.0.3. Parameter bounds were set to: $k_f$ = 0.1–0.7, $k_s$ = 0–0.01, and $f_f$ = 0–0.2, following Schädel et al. (Schädel, Luo, David Evans, Fei, & Schaeffer, 2013). A detailed description about the optimization approach can be found in Zhang et al. (2024).

EXP.1. Best-fit model (full optimization). All model parameters – $k_f$, $k_s$, and $f_f$ – were optimized for each temperature treatment in each incubation trial. However, in one trial, the same $f_f$ value was shared across different incubation temperatures. This represents the best-case model performance and serves as the baseline for comparison.

EXP.2. Inherent temperature response. We fixed the decomposition rates $k_f$ and $k_s$ and pool size ($f$) to those optimized at the lowest incubation temperature in EXP.1 and applied a single optimized $Q_{10}$ value to scale the rates across other temperatures. The response of a specific SOC pool to temperature depends on its chemical decomposability and the thermal traits of the
associated microbial community. Forcing the temperature sensitivities to be the same (i.e., a single $Q_{10}$) across carbon pools effectively eliminates these distinct responses, thereby isolating the effect of microbial and substrate-related intrinsic temperature response.

EXP.3. External environmental constraints. We again used the $k$ values at the lowest incubation temperature and $f$ values from EXP.1. However, here we applied globally averaged $Q_{10}$ values (derived from the empirical dataset) uniformly across all
samples. This constrains all soils to a common temperature response, allowing us to quantify the effect of site-specific external constraints on microbial access to SOC.

To evaluate the contribution of each mechanism to model performance, we calculated their marginal importance relative to the best-fit model following Eq.(4-8) (Grömping, 2007):

$$I_1 = R^2_{EXP.1} - R^2_{EXP.2} \ (inherent \ response) \ , \tag{4}$$

$$I_2 = R^2_{EXP.2} - R^2_{EXP.3} \ (external \ constraints) \ , \tag{5}$$

$$P_1 = \frac{I_1}{I_1 + I_2} \cdot R^2_{EXP.1} \ , \tag{6}$$

$$P_2 = \frac{I_2}{I_1 + I_2} \cdot R^2_{EXP.1} \ , \tag{7}$$

$$Unexplained = 1 - R^2_{EXP.1} \ , \tag{8}$$

where $I_1$ and $I_2$ represent the importance of inherent temperature response and external environmental constraints, respectively;
$P_1$ and $P_2$ denote the relative importance of two mechanisms, respectively; $Unexplained$ indicate the unexplained portion of total variation. We applied bootstrap resampling (n = 5,000) to estimate the mean and 95% confidence interval for the relative importance of each mechanism across all incubation trials.

## 5.3 Simulation results

The modeling experiments revealed distinct contributions of intrinsic and extrinsic mechanisms to the temperature sensitivity
of SOC mineralization. The model explained, on average, 80%, 71%, and 61% of the variance in SOC mineralization for EXP.1, EXP.2, and EXP.3, respectively (Fig. 5A). RMSE increased accordingly across the three experiments (Fig. S2). Relative to EXP.1, model performance in EXP.2 showed a decline ranging from +0.1% to –62.6% with an average of –11.1%.

Relative to EXP.2, model performance in EXP.3 exhibited changes from +18.3% to -99.9% with an average of –15.4% (Fig.
5B). Overall, intrinsic temperature response and extrinsic environmental constraints contributed comparably at the global scale,
with intrinsic response accounting for 41% with a 95% confidence interval (CI) of 38%–43% and environmental constraints
contributing 39% (95% CI: 37%–42%) to the total variance (Fig. 5C). However, substantial variation emerged across
ecosystems. In croplands, intrinsic temperature response was dominant, contributing 50% (95% CI: 45%–54%), whereas
environmental constraints accounted for a smaller share (33% with the 95% CI of 28%–37%). In contrast, wetlands exhibited
the opposite pattern, with environmental constraints contributing 52% (95% CI: 44%–61%) and intrinsic temperature response
contributing only 30% (95% CI: 22%–38%).

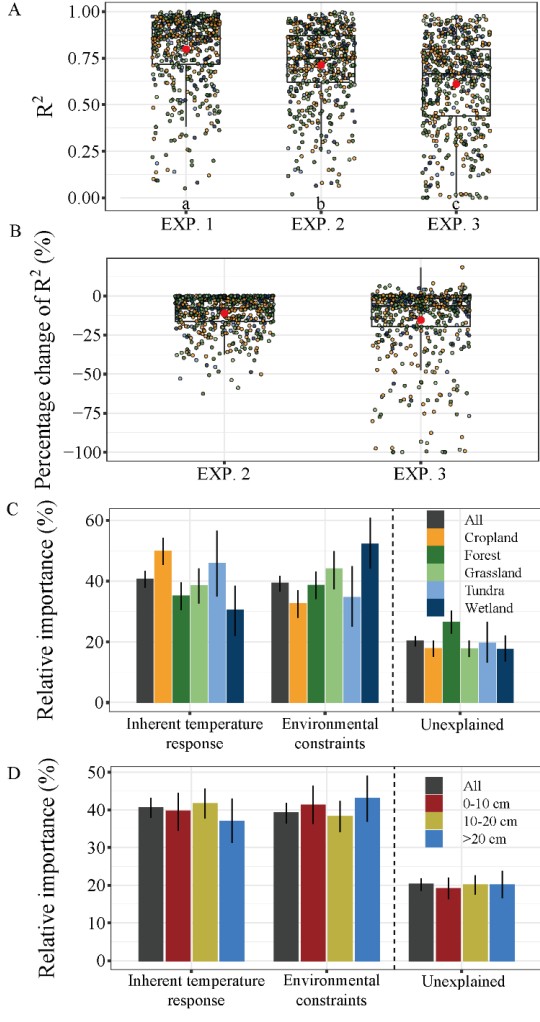

**Figure 5: The relative importance of intrinsic temperature responses and external environmental constraints.** Panels A and B show
the determination coefficients ($R^2$) and the corresponding percentage changes in $R^2$, respectively, for the three simulation experiments.
Panels C and D represent relative importance of the two mechanisms categorized by ecosystem type and soil depth, respectively. B, the
percentage change in $R^2$ of EXP. 2 relative to EXP. 1, and the percentage change in $R^2$ of EXP. 3 relative to EXP. 2. Error bars in panels C
and D represent the 95% confidence intervals based on 5,000 bootstrap resamples of the original relative importance. EXP. 1 represents the



best-fit model and serves as the baseline for comparison, EXP. 2 aims to assess the relative importance of the intrinsic temperature response, and EXP. 3 aims to assess the relative importance of external environmental constraints. Different letters in panel A indicate significant difference ($p<0.05$).

These contrasting patterns reflect ecosystem-specific controls on the temperature sensitivity of SOC mineralization. In croplands, frequent soil disturbances such as tillage, fertilization, and residue management likely enhance substrate availability and microbial activity, thereby amplifying the role of intrinsic biological and chemical processes (Chen et al., 2019). In wetlands, by contrast, saturated conditions impose strong oxygen limitations and redox constraints on microbial activity, making abiotic environmental factors the primary regulator of SOC turnover (Chen, Zou, Cui, Nie, & Fang, 2018). These

findings underscore the importance of incorporating ecosystem-specific mechanisms into ESMs – particularly in systems shaped by hydrological regimes or intensive management – is critical for improving projections of soil carbon-climate feedbacks under global warming.

     There were no significant differences of the relative importance of the two regulatory mechanisms across soil depths (Fig. 5D). However, it should be noted that subsoil layers was underrepresented, particularly for layers between 0.5 m (Fig. 2C).

Together, our results highlight the importance of integrating both intrinsic and extrinsic mechanisms into understanding temperature response functions.

     However, it is critical to acknowledge that the "intrinsic temperature response" in our modelling framework encompasses both SOC chemical decomposability and microbial metabolic activity, as these processes are inherently intertwined in carbon turnover (Conant et al., 2011). For example, an increase in the decomposition rate constant ($k$) with temperature could reflect

enhanced microbial enzyme kinetics, but may also be driven by temperature-induced changes in substrate availability via increased diffusion or depolymerization of complex carbon compounds (Conant et al., 2011). Similarly, shifts in the fraction of fast-cycling carbon ($f_f$) may not solely indicate a change in carbon pool composition, but also microbial substrate preferences or physiological adjustments that alter carbon allocation between biomass production and respiration (Zheng et al., 2025). These caveats underscore the need for more detailed, trait-explicit models that separately track microbial physiology, substrate

quality, and abiotic accessibility (Zhang et al., 2024).

## 6 CONCLUSIONS AND FUTURE VISION

SOC dynamics are central to predicting terrestrial carbon–climate feedbacks, yet remain a major source of uncertainty in ESMs. By synthesizing a comprehensive global dataset of SOC mineralization under controlled incubation conditions, this study provides a robust framework to evaluate the temperature sensitivity of SOC decomposition and the mechanisms that govern

it. Our findings highlight that external environmental constraints—such as physicochemical protection and substrate accessibility—and intrinsic SOC decomposability play similarly important roles in shaping temperature responses, but their relative influence is ecosystem-dependent. Moreover, we demonstrate that widely used temperature response functions in ESMs often fail to capture observed patterns, particularly under temperature extremes or in specific ecosystems.

     Based on our analyses, we propose following priorities for advancing SOC-climate research:



1)    Expand spatial and vertical coverage of soil sampling

Despite the growing number of incubation studies, current datasets remain heavily biased toward surface soils, mid-latitude systems, and short-term incubations. Particularly underrepresented are data from extreme environments (e.g., tundra, wetlands, deserts), subsoil layers, and high or low incubation temperatures—all of which are crucial for understanding carbon–climate feedbacks in vulnerable or carbon-dense regions. Addressing these gaps through targeted sampling campaigns and

standardized data collection would enhance model calibration, validation, and transferability across scales.

2)    Align incubation design with ecologically relevant temperature scenarios

Laboratory incubation conditions – although ideal for isolating mechanisms – may fail to replicate the complexity of natural systems. Field conditions introduce fluctuating moisture regimes, plant-microbe interactions, freeze-thaw cycles, and other dynamic processes that strongly mediate SOC responses. We advocate for hybrid approaches that combine laboratory

incubation data with in situ measurements (e.g., eddy covariance fluxes, carbon isotope tracing) and long-term warming experiments to ground-truth model behaviour and improve ecological relevance.

3)    Integrate mechanistic constraints into models

Most SOC temperature response functions currently used in ESMs are based on simplified relationships that fail to incorporate critical regulatory mechanisms. Our findings clearly demonstrate that these simplified functions often

underperform when applied to real-world data, particularly across diverse ecosystems and temperature regimes. Embedding mechanistic constraints, such as mineral protection, oxygen limitation, and depth-specific carbon turnover, into temperature response formulations (Bradford et al., 2016) could substantially improve the fidelity of SOC projections under future climate scenarios.

4)    Advance spatial scaling

Most ESMs still apply uniform temperature response functions across broad geographic regions, neglecting site-specific variability in soil properties, mineralogy, hydrology, and microbial ecology. Our findings argue for a more spatially explicit representation of SOC temperature responses. Advances in machine learning, data assimilation, and remote sensing provide promising tools for spatial upscaling of temperature response parameters, enabling site-specific calibration of ESMs. Integrating knowledge-guided machine learning with mechanistic soil biogeochemistry models (Liu et al., 2024) would

significantly enhance predictive accuracy and reduce uncertainty in regional and global carbon-climate feedback estimates.

Together, these priorities call for a more mechanistic, depth-aware, and spatially explicit framework for investigating SOC mineralization. By coupling empirical datasets with process-based modelling and machine learning, the soil carbon research community can significantly reduce uncertainty in carbon–climate feedbacks and improve projections of SOC stability in a warming world.



**ACKNOWLEDGMENTS:** This study is financially supported by the National Natural Science Foundation of China (grant no. 32171639) and the China National Postdoctoral Program (grant no. BX20240324).

**CONFLICT OF INTEREST:** The authors declare no competing financial interests.

**AUTHORSHIP:** Z.L. conceived the study; S.Z. led data collection, assessed the data, and wrote the first draft; Z.L. revised the manuscript and led results interpretation with the contribution of all authors.

**DATA AVAILABILITY STATEMENT:** The data that support the findings of this study are available in Figshare at: https://doi.org/10.6084/m9.figshare.25808698 (Zhang et al., 2025).

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
