# Peer review of "A global dataset of soil organic carbon mineralization in response to incubation temperature changes"

_Earth System Science Data, 2025_

## Author Comment (AC1)

**Response to referee #2**

**Referee #2:**

**Comments:**

[Comment 1] This study compiles a global dataset of 22,000 observations from 191 incubation experiments on soil organic carbon (SOC) mineralization to assess how temperature controls soil CO2 release, a key climate—carbon feedback. The dataset, largely biased toward surface soils, short incubations, and mid-latitude regions, reveals major gaps in deep soils, extreme ecosystems, and Africa. The author suggest that Earth System Models generally misrepresent SOC temperature sensitivity, especially under warming extremes, though multi-term exponential functions perform best. Using a two-pool carbon model, the authors find that both intrinsic factors (SOC quality, microbial traits) and extrinsic constraints (oxygen, mineral protection, moisture) contribute equally to global SOC responses, but their relative importance varies by ecosystem (e.g., croplands vs. wetlands). The work highlights the urgent need for more representative experiments and improved model formulations to reduce uncertainty in carbon—climate feedback projections. Although I find the article interesting and the dataset very useful, I have a number of comments that could help improve the readability of the manuscript.

**Response:** We thank the reviewer for the positive comments and constructive suggestions. We have carefully revised the manuscript to improve its readability and clarity.

[Comment 2] Firstly, the mathematical approach used to distinguish between internal and external effects is rather poorly explained in section 5.4, which makes section 5.5 more difficult to understand. I suggest that the authors provide more detail in section 5.4 and perhaps simplify the vocabulary used.

**Response:** Thank the reviewer for pointing this out. We have thoroughly revised sections 5.4 and 5.5 to clarify the mathematical approach used to distinguish intrinsic from extrinsic effects (lines 276–296). We have also simplified the terminology to enhance readability and aid comprehension.

[Comment 3] It is noted in several places that you tested the temperature response functions of 69 ESMs. This is inaccurate, as most of the models whose temperature response functions you tested are not ESMs. This approximation should therefore be corrected throughout the manuscript.

**Response:** We thank the reviewer for highlighting this inaccuracy. We have replaced the term "ESMs" with "land surface models" throughout the manuscript.

[Comment 4] Table 2 also needs to be simplified because some of the information is incorrect. For example, for Jules, the centre is described as the UK, which is a country, whereas for the other models, the authors give a research group instead.

**Response:** The "Modeling centre" column has been removed from Table 2, as it contained inconsistent information and was not essential to the table's purpose.

[Comment 5] CENRUTY-> CENTURY

**Response:** Thank the reviewer for the careful review. Corrected.

[Comment 6] It is also unclear in this table what the difference is between 'land carbon' and 'land surface models'.

**Response:** Thank the reviewer for the careful review. In our study, land surface models refer to the full terrestrial components of Earth System Models that simulate energy, water, and carbon exchanges, whereas land carbon models are submodules that specifically represent terrestrial carbon cycling processes (e.g., photosynthesis, respiration, soil carbon decomposition). We have clarified this distinction in the revised table (lines 188–190).

[Comment 7] Figure 4 also needs improvement as it lacks clarity, particularly as M1 and M2 have not been defined.

**Response:** We thank the reviewer for this helpful comment. We have now defined M1 and M2 in the figure caption (line 251–254) and revised Figure 4 to improve clarity and readability.

[Comment 8] How eq. 3 affect eq. 2 in the model developed by the authors?

**Response:** In our simulations, model parameters were optimized for each incubation trial using Eq. (2) by comparing modeled and observed SOC mineralization rates across multiple incubation temperatures. Specifically, the decomposition rate constants ( $k_f$  and  $k_s$ ) and their temperature sensitivities ( $Q_{10\_fast}$  and  $Q_{10\_slow}$ ) were first optimized at the lowest incubation temperature. The corresponding k values at higher temperatures were then scaled using Eq. (3) based on the optimized  $Q_{10}$  values.

[Comment 9] Section 2 L. 78 point 2) More details are needed here, for instance do you accept when the same samples were incubated at 2 different temperatures?

**Response:** Yes, we included data from studies where the same soil samples were incubated under two or more different temperatures. This was clarified in point 2: *Each experiment must incubate the same soil at two or more temperatures*. (line 80)

[Comment 10] Do you use equal time or equal C (Hamdi et al., 2012)?

Hamdi, S., Moyano, F., Sall, S., Bernoux, M., Chevallier, T., 2012. Synthesis analysis of the temperature sensitivity of soil respiration from laboratory studies in relation to incubation methods and soil conditions. Soil Biol Biochem 58, 115–126. https://doi.org/10.1016/j.soilbio.2012.11.012

**Response:** In our case study, we used the equal-time method (Eq. 1) to estimate  $Q_{10}$ . This information has been added to the manuscript (lines 66 and 205).

[Comment 10] L323: "There were no significant differences of the relative importance..." how this was tested?

**Response:** We thank the reviewer for this question. We conducted pairwise significance tests using a bootstrap approach (5,000 resamples) to assess differences in the relative importance of each mechanism. Specifically, for each mechanism, we resampled the simulated relative importance values derived from independent model

runs. For each bootstrap iteration, we computed the mean difference between two ecosystems or soil depths, and generated an empirical distribution of these differences under the null hypothesis of no difference. Statistical significance (p < 0.05) was determined based on whether the 95% CI of the difference excluded zero. The results have been updated in the revised Fig. 5, and the statistical method has been described in the **5.2 Simulation experiments** section (lines 307-310).

---

## Author Comment (AC2)

**Response to referee #1**

**Referee #1:**

**Comments:**

[Comment 1] Nominally (see below), this manuscript describes a dataset of soil incubations focusing on carbon mineralization and temperature sensitivity (Q10) calculation. This is interesting and important for reasons well laid out in the introduction, as such incubations have been a major source of information about this process and informed models and understanding at many scales; an analysis-ready dataset of incubations is valuable. The authors' dataset is publicly posted, has almost 22,000 rows, and seems clearly laid out (although see #2 below).

**Response:** We thank the reviewer for the thoughtful review and are grateful for these positive comments.

[Comment 2] That said, there are several significant problems here. First, the ms is oddly structured. It essentially has three parts: (i) a description of the dataset; (ii) data summaries and comparison with ancillary data (in particular, incubation temperatures compared with the mean annual temperature of sampling location); and, very unexpectedly, (iii) an extended summary of earth system model approaches to decomposition and simple modeling exercise involving the dataset. From https://www.earth-system-science-data.net/about/aims\_and\_scope.html, the scope of ESSD is "Articles in the data section may pertain to the planning, instrumentation, and execution of experiments or collection of data. Any interpretation of data is outside the scope of regular articles." Based on this, I think that (iii) above is clearly out of scope; it's extremely odd to find this ESM algorithm analysis in an ESSD ms, and it should be removed. Even (ii) strikes me as marginal in terms of scope—it's analysis, not data description!

**Response:** We appreciate the reviewer's concern regarding the alignment of our manuscript with the scope of ESSD. Our intention with point (ii) was not to perform independent analyses, but rather to highlight gaps between laboratory incubation settings and real-world conditions. Identifying these gaps provides important context for users, helps clarify potential limitations when applying the dataset, and offers practical recommendations for designing more realistic future experiments. We believe this enhances the utility of the dataset rather than extending beyond ESSD's scope.

Regarding point (iii), our simple modeling exercises are presented as case studies to demonstrate how the dataset can be applied, particularly with models. These are not meant as stand-alone scientific interpretations, but as illustrations of data use in model frameworks. Such case studies are common in ESSD papers, where they help maximize dataset impact. For example, Schädel et al. (2020) combined SIDb with five carbon models to compare model performance and to illustrate the dataset's utility for constraining model representations of soil carbon turnover. Ménard et al. (2019) explicitly compared their meteorological dataset against snow model outputs to illustrate utility for model benchmarking, while Hong et al. (2022) used their dataset to analyze terrestrial surface temperature trends from 2003 to 2019. These studies

demonstrate that data-model integration is within ESSD's practice when it illustrates dataset utility.

To address the reviewer's concerns, we have expanded relevant clarifications in the ending paragraph of the Introduction and in the section of "Simulation results" to point out that these analyses are presented as demonstrations of dataset applicability: "To showcase the dataset's utility and scientific potential, we used it in a soil carbon model as a case study. This analysis demonstrates its applicability to process-based modeling and its contribution to understanding soil carbon dynamics" (lines 63-65); "These case study results underscore the potential of the dataset for facilitating model-data integration, exploring the mechanisms underlying SOC dynamics in response to climate change, and refining model representations under future warming." (lines 344-346).

**References:**

Schädel, C., Beem-Miller, J., Aziz Rad, M., Crow, S. E., Hicks Pries, C. E., Ernakovich, J., Hoyt, A. M., Plante, A., Stoner, S., Treat, C. C., and Sierra, C. A.: Decomposability of soil organic matter over time: the Soil Incubation Database (SIDb, version 1.0) and guidance for incubation procedures, Earth Syst. Sci. Data, 12, 1511–1524, https://doi.org/10.5194/essd-12-1511-2020, 2020.

Ménard, C. B., Essery, R., Barr, A., Bartlett, P., Derry, J., Dumont, M., Fierz, C., Kim, H., Kontu, A., Lejeune, Y., Marks, D., Niwano, M., Raleigh, M., Wang, L., and Wever, N.: Meteorological and evaluation datasets for snow modelling at 10 reference sites: description of in situ and bias-corrected reanalysis data, Earth Syst. Sci. Data, 11, 865–880, https://doi.org/10.5194/essd-11-865-2019, 2019.

Hong, F., Zhan, W., Göttsche, F.-M., Liu, Z., Dong, P., Fu, H., Huang, F., and Zhang, X.: A global dataset of spatiotemporally seamless daily mean land surface temperatures: generation, validation, and analysis, Earth Syst. Sci. Data, 14, 3091–3113, https://doi.org/10.5194/essd-14-3091-2022, 2022.

**[Comment 3]** Second, there's no mention of SIDb (https://soilbgc-datashare.github.io/sidb/). The SIDb paper (Schädel et al. 2020) is cited but it's bizarre not to note and discuss \*at length\* this pre-existing and seemingly very similar effort. How much overlap is there between the authors' work and SIDb? Why not contribute these data to SIDb, rather than duplicate work and confuse researchers?

**Response:** We agree that SIDb (Schädel et al. 2020) is an important and valuable dataset, and we have cited it in our main manuscript. However, our dataset differs in focus and structure:

- Scientific scope. SIDb dataset compiles data to track soil carbon mineralization dynamics over time, whereas our dataset is specifically designed around experiments where the same soil was incubated at two or more temperatures. This structure allows direct assessment of temperature sensitivity, which is the central focus of our study.
- 2. Coverage. SIDb includes 31 studies, 11 of which overlap with ours. Our dataset incorporates 192 studies, substantially expanding the scope.

3. Auxiliary information. Our dataset includes extensive experimental details (e.g., sieving, pretreatment, soil moisture, soil mass), soil profile information, , and site characteristics (e.g., vegetation species, coordinates). These additional variables are not available for SIDB and increase the dataset's valuable for model calibration and mechanistic assessment of soil carbon dynamics.

For these reasons, we see the two datasets as complementary rather than duplicative. We believe that maintaining ours as a stand-alone, thematically focused dataset maximizes visibility and usability for the research community.

In addition, to address the reviewer's concern and clarify the novelty of our study, we have revised the title from "A global dataset of soil organic carbon mineralization under various incubation conditions" to "A global dataset of soil organic carbon mineralization in response to incubation temperature changes." This modification underscores that our work specifically targets the temperature response of soil carbon mineralization, distinguishing it clearly from SIDb focusing on broader incubation conditions.

[Comment 4] Finally, as already noted I have concerns about the structure of the data and how it doesn't support easy reproducibility in terms of finding the source studies.

**Response:** Thank you for this point. Please refer to the response to Comment 7 for clarification on how reproducibility and traceability of source studies are ensured.

[Comment 5] In summary, while I appreciate the large amount of work here, and believe this dataset will be valuable, the current ms should be rejected or subject to fundamental revisions.

**Response:** We thank the reviewer for recognizing the value of our dataset. We have carefully revised the manuscript, clarified scope issues, and improved transparency to address the concerns raised. We believe these clarifications and revisions substantially strengthen the manuscript and bring it in line with ESSD's expectations.

**Specific comments:**

[Comment 6] Line 35: "expressed as"

**Response:** Thank you for the careful review. Corrected.

[Comment 7] The dataset structure as posted on Figshare is a little odd. The study information is combined with the observational data, i.e. it's all in a single CSV file, so many rows are duplicated; having separate "data" and "studies" files might be clearer and cleaner. In addition, there's no DOI, URL, or volume/issue information...to find a paper, are users supposed to search the title? Having a machine-searchable link or DOI seems crucial.

**Response:** We appreciate the reviewer's suggestion regarding dataset structure. Our initial decision to combine all data in a single CSV file was deliberate, as it allowed researchers to access metadata and measurements in one place, facilitating analysis without repeatedly merging multiple files. Although this approach introduces some repeated metadata entries, it is a common practice in large datasets and does not hinger usability when the metadata are well organized. This structure also enables flexible querying. For example, users can easily filter carbon mineralization rates for a specific

study, soil, or incubation temperatures.

That said, we agree that separating study-level and data-level information will improve clarity and reproducibility. In the revised version, we now provide two files: *data.csv*, which contains only observational data, and *studies.csv*, which contains study-level metadata. We have also added DOIs for all studies in *studies.csv* to ensure that sources are easily identifiable and machine-searchable.

[Comment 8] 492-498: duplicated reference

Response: Corrected.

---

## Author Response (AR2)

**Response to referees**

**Referee #2:**

**Comments:**

**[Comment 1]** The revised version is much more clear but I think the term land surface model is not used correctly in particular in the table 2 where some models are not land surface models (RothC, Century for instance). The same comment applied to the conclusion. I suggest to use a more generic term.

**Response:** We thank the reviewer for pointing this out. To improve accuracy, we have revised the term at its first occurrence to "land surface and/or carbon models (hereafter referred to as carbon models)" (lines 65-66). Accordingly, we now use carbon models consistently throughout the manuscript, including in Table 2 and the Conclusion section.